# Neurodegeneration Biomarkers in Adult Spinal Muscular Atrophy (SMA) Patients Treated with Nusinersen

**DOI:** 10.3390/ijms25073810

**Published:** 2024-03-29

**Authors:** Pol Andrés-Benito, Juan Francisco Vázquez-Costa, Nancy Carolina Ñungo Garzón, María J. Colomina, Carla Marco, Laura González, Cristina Terrafeta, Raúl Domínguez, Isidro Ferrer, Mónica Povedano

**Affiliations:** 1Neurologic Diseases and Neurogenetics Group, Institute of Biomedical Research (IDIBELL), 08907 Barcelona, Spain; 2CIBERNED (Network Centre of Biomedical Research of Neurodegenerative Diseases), Institute of Health Carlos III, 08907 Barcelona, Spain; 3Neuromuscular Unit and ERN-NMD Group, Department of Neurology, Hospital Universitario y Politécnico La Fe and IIS La Fe, Centro de Investigación Biomédica en Red de Enfermedades Raras (CIBERER), 46026 Valencia, Spain; 4Department of Medicine, University of Valencia, 46021 Valencia, Spain; 5Anesthesia and Critical Care Department, Bellvitge University Hospital-University of Barcelona, 08907 Barcelona, Spain; 6Functional Unit of Amyotrophic Lateral Sclerosis (UFELA), Department of Neurology, Bellvitge University Hospital, 08907 Barcelona, Spain; 7Neuropathology Group, Institute of Biomedical Research (IDIBELL), 08907 Barcelona, Spain; 8Department of Pathology and Experimental Therapeutics, University of Barcelona, 08907 Barcelona, Spain

**Keywords:** adult spinal muscular atrophy, nusinersen treatment, biomarkers

## Abstract

The objective of this study is to evaluate biomarkers for neurodegenerative disorders in adult SMA patients and their potential for monitoring the response to nusinersen. Biomarkers for neurodegenerative disorders were assessed in plasma and CSF samples obtained from a total of 30 healthy older adult controls and 31 patients with adult SMA type 2 and 3. The samples were collected before and during nusinersen treatment at various time points, approximately at 2, 6, 10, and 22 months. Using ELISA technology, the levels of total tau, pNF-H, NF-L, sAPPβ, Aβ40, Aβ42, and YKL-40 were evaluated in CSF samples. Additionally, plasma samples were used to measure NF-L and total tau levels using SIMOA technology. SMA patients showed improvements in clinical outcomes after nusinersen treatment, which were statistically significant only in walkers, in RULM (*p* = 0.04) and HFMSE (*p* = 0.05) at 24 months. A reduction in sAPPβ levels was found after nusinersen treatment, but these levels did not correlate with clinical outcomes. Other neurodegeneration biomarkers (NF-L, pNF-H, total tau, YKL-40, Aβ40, and Aβ42) were not found consistently changed with nusinersen treatment. The slow progression rate and mild treatment response of adult SMA types 2 and 3 may not lead to detectable changes in common markers of axonal degradation, inflammation, or neurodegeneration, since it does not involve large pools of damaged neurons as observed in pediatric forms. However, changes in biomarkers associated with the APP processing pathway might be linked to treatment administration. Further studies are warranted to better understand these findings.

## 1. Introduction

Spinal muscular atrophy 5q (SMA) is a rare autosomal recessive inherited disorder characterized by the degeneration of α-motor neurons of the spinal cord and brainstem. This results in proximal spinal and bulbar muscle weakness and atrophy, often affecting respiratory muscles and leading to respiratory failure, reducing the life expectancy [1,2]. SMA is caused by bi-allelic loss-of-function through the deletion, conversion, or mutation of the survival motor neuron 1 gene (SMN1), located on chromosome 5q13, coding for the survival motor neuron protein, which is particularly important for the functioning of α-motor neurons. In addition to the SMN1 gene, humans have one or more paralogous SMN1 gene copies, named the survival motor neuron 2 gene (SMN2), which only differ by a few bases and also produce the protein, although at much lower levels (approximately 10%) [3,4]. Thus, the lethality that would occur due to the complete absence of the SMN protein in SMN1 mutated forms is partially rescued by the presence of a variable number of copies of the paralogous gene. Therefore, extra copies of the nearby related gene may modify the severity of SMA [3,4].

Classically, four types of SMA have been differentiated depending on the age at onset and the maximum milestone reached. SMA type 1 (SMA1) is the most severe form of SMA. It accounts for 50–70% of cases of childhood-onset SMA, and it is characterized by the onset of symptoms before the age of six months, as well as a median life expectancy of less than two years. Children with SMA type 2 (SMA2) have a milder form of the disease, with symptoms starting between the 6th and 18th months of age. These infants can sit without support, and a few of them can stand with leg braces, but none can walk independently. Difficulty in swallowing and coughing and respiratory insufficiency may occur during adolescence and become frequent during adulthood. Patients with SMA type 3 (SMA3) have disease onset after 18 months. They will be able to walk independently, although some require wheelchair assistance later in life. In patients with SMA type 4 (SMA4) symptoms usually start after the third or fourth decade of life, and patients suffer from milder motor impairment and usually will not lose ambulation [1,2,3,4].

Since the discovery of the main disease-causing gene (SMN1), substantial progress in understanding the molecular pathogenesis of this disease is ongoing, developing disease-modifying therapies, aimed at increasing the amount of the survival motor neuron protein. For example, the antisense oligonucleotide drug nusinersen administered intrathecally was approved for the treatment of pediatric and adult SMA. Clinical studies showed an improvement in motor function in children with SMA type 1 and 2 [5,6,7,8]. After the approval of nusinersen, real-world studies have also shown milder improvements in a subset of adult patients with SMA types 2, 3, and 4 [9,10,11,12]. However, other patients remain stable or deteriorate in motor scales, and it is unclear if these patients are benefiting from treatment or not, because motor scales show several limitations for the assessment of adult SMA patients [10]. A variety of biomarkers have been investigated for their potential use in the diagnosis and monitoring of neurodegenerative diseases [13,14,15,16,17,18,19,20]; however, the role of these putative biomarkers for predicting or monitoring the response to nusinersen in adult SMA patients is not well known. We hypothesized that specific biomarkers linked to target pathways in neurodegenerative diseases might be useful to assess target engagement in each adult SMA patient and predict the nusinersen treatment response. The present study only includes nusinersen as an adult SMA treatment because at the time of its design and approval (2020), it was the only drug approved at the state level in Spain by the “Spanish Agency of Medicines and Medical Devices” for the treatment of adult SMA.

## 2. Results

Thirty-one adults with SMA were included in this study. Samples used in this study belong to 19 men and 12 women suffering from adult SMA at different nusinersen dose time treatment and 12 men and 18 women controls without neurological diseases or alterations. It is important to note that, due to the difficulty of obtaining cerebrospinal fluid (CSF) from control young age-matched healthy patients, we used control samples obtained from the Traumatology Unit of the Bellvitge Hospital, from patients with trauma and breakage, that are usually older (69.96 ± 1.98 years) than adult SMA patients (37.3 ± 2.20 years) (*p* < 0.001). Seven SMA patients did not complete the entire follow-up period (by the patient’s decision) and were therefore excluded from the clinical outcome analysis and from the biomarker correlations with clinical data (*n* = 24). CSF and plasma hemolyzed samples were discarded for biomarker analysis. Thus, the present study includes the assessment of neurodegeneration biomarkers in 130 samples of CSF and 133 samples of plasma belonging to 31 patients with adult SMA treated with the nusinersen drug at five treatment dose times, in addition to 30 CSF and plasma samples from healthy older adult patients as controls. Table 1 summarizes the baseline characteristics of the patients and the number of cases used in this study.

### 2.1. Nusinersen Leads to Clinical Improvement after 24 Months of Treatment

The progression of clinical variables over time in the 24 patients who completed the 24-month evaluation is described in Table 2 and graphically represented (Figure 1), illustrating the mean values. Although no statistically significant differences between scores in the whole adult SMA patient cohort were found, there was an overall tendency towards an improvement in clinical outcomes when comparing the mean values of each group (Table 2). When patients were categorized as “non-sitters”, “sitters”, and “walkers”, an appreciable improvement was observed, compared to the baseline values, in the Hammersmith functional motor scale (HFMSE) and the revised upper limb module (RULM) clinical scales among “walkers” and in the Revised Amyotrophic Lateral Sclerosis Functional Rating Scale (ALSFRS-R) among “sitters” at 12 months (*p* = 0.05) but not in “non-sitter” patients (Figure 1). Specifically, this improvement in “walker” patients was statistically significant in the HFMSE at 12 months (*p* = 0.047) and 24 months (*p* = 0.05) and in the RULM at 12 months (*p* = 0.04). No other statistically significant improvements were found.

### 2.2. Neurofilament Levels in CSF and Plasma of Adult SMA Patients Treated with Nusinersen

Lower CSF neurofilament light chain (NF-L) levels were found in samples of adult SMA patients before nusinersen treatment (SMA(Pre): 485.38 ± 95.69 pg/mL) compared to the healthy control group (control: 1004.30 ± 95.69 pg/mL). However, after adjusting for age and sex in the multivariable model, this difference was not statistically significant [F(1,56) = 2.123, *p* = 0.151]. Moreover, this analysis revealed that older age [F(1,56) = 4.587, *p* = 0.037] and male sex [F(1,56) = 10.439, *p* = 0.002] are associated with higher NF-L levels (Figure 2A).

Furthermore, a one-way ANOVA of repeated measures was conducted to examine the differences between samples of adult SMA patients before nusinersen treatment administration (SMA(Pre): 485.38 ± 95.69 pg/mL) and samples after nusinersen treatment administration at different time points: 2 months (SMA(2 M): 570.14 ± 75.70 pg/mL), 6 months (SMA(6 M): 514.96 ± 55.46 pg/mL), 10 months (SMA(10 M): 608.48 ± 81.45 pg/mL), and 22 months (SMA(22 M): 511.27 ± 44.99 pg/mL) (Figure 2A). The results of the ANOVA did not indicate significant differences in CSF NF-L levels at the different time points [F(2.975,64.71) = 0.8282, *p* = 0.4823] (Figure 2A).

In parallel, we also analyzed NF-L levels in plasma samples using SIMOA technology in our patient cohort. In this case, the linear regression model adjusted for age and sex did not show significant differences in plasma NF-L levels between the healthy control samples (control: 13.93 ± 1.34 pg/mL) and the non-treated adult SMA patient samples (SMA(Pre): 10.37 ± 1.11 pg/mL) due to the group effect [F(1,55) = 0.481, *p* = 0.491]. Additionally, this analysis suggested that age [F(1,55) = 3.866, *p* = 0.055] tended to increase NF-L levels in plasma, while sex as a covariate [F(1,55) = 0.029, *p* = 0.866] did not show a significant association with changes in plasma NF-L levels (Figure 2B).

Next, we performed a one-way ANOVA test of repeated measures to examine changes in NF-L levels in plasma across different time doses of nusinersen-treated adult SMA patients. The statistical analysis revealed a significant effect of treatment administration [F(2.741,61.67) = 4.527, *p* = 0.0077] on NF-L plasma levels when comparing samples of adult SMA patients treated at 2 months (SMA(2 M): 11.48 ± 1.16 pg/mL) with samples of adult SMA patients treated at different time doses: 6 months (SMA(6 M): 9.01 ± 1.13 pg/mL, *p* = 0.01), 10 months (SMA(10 M): 9.41 ± 0.81 pg/mL, *p* = 0.05), and 22 months (SMA(22 M): 9.33 ± 0.81 pg/mL, *p* = 0.022). However, no significant difference was observed when comparing samples from different time points with the baseline levels of samples from non-treated adult SMA patients (SMA(Pre): 10.37 ± 1.11 pg/mL) (Figure 2B).

Phosphorylated neurofilament heavy chain (pNF-H) levels in CSF exhibited a high degree of variability among patient groups. The linear regression model, adjusted for age and sex, revealed nonsignificant differences in plasma pNF-H levels between samples of the control group (control: 0.54 ± 0.10 ng/mL) and samples of non-treated adult SMA patients (SMA(Pre): 0.24 ± 0.06 ng/mL) [F(1,57) = 0.273, *p* = 0.603]. However, significant changes were associated with sex as a covariate [F(1,57) = 4.752, *p* = 0.034] but not with age as a covariate [F(1,57) = 2.609, *p* = 0.112] (Figure 2C).

A subsequent repeated measures one-way ANOVA revealed significant differences between pNF-H CSF levels in non-treated adult SMA patients (SMA(Pre): 0.24 ± 0.06 ng/mL) and CSF samples from treated adult SMA patients at different time doses [F(2.571,57.85) = 3.111, *p* = 0.04]. Specifically, significant increases were observed between samples of non-treated adult SMA patients (SMA(Pre): 0.24 ± 0.06 ng/mL) and CSF samples of adult SMA patients treated at 10 months (SMA(10 M): 0.57 ± 0.09 ng/mL, *p* = 0.0083). Additionally, significant differences were found between CSF samples of adult SMA patients treated at 10 months (SMA(10 M): 0.57 ± 0.09 ng/mL) and CSF samples of adult SMA patients treated at 22 months (SMA(22 M): 0.32 ± 0.08 ng/mL, *p* = 0.0292) (Figure 2C).

### 2.3. Total Tau Levels in CSF and Plasma of Adult SMA Patients Treated with Nusinersen

The linear regression model, adjusted for age and sex, revealed significantly increased levels of total tau in CSF samples from non-treated SMA patients (SMA(Pre): 181.29 ± 12.98 pg/mL) compared to CSF samples from healthy controls (control: 176.21 ± 15.55 pg/mL) due to the group effect [F(1,58) = 11.366, *p* = 0.0012]. Moreover, these changes were also found to be associated with age [F(1,58) = 16.522, *p* = 0.000] but not with sex [F(1,58) = 0.833, *p* = 0.352] (Figure 3A).

A subsequent repeated measures one-way ANOVA did not reveal significant differences in total tau levels in CSF samples before and during nusinersen treatment [F(2.7,61.43) = 1.056, *p* = 0.37]: pretreatment (SMA(Pre): 181.29 ± 12.98 pg/mL), 2 months (SMA(2 M): 177.25 ± 13.97 pg/mL), 6 months (SMA(6 M): 175.93 ± 18.27 pg/mL), 10 months (SMA(10 M): 155.60 ± 18.52 pg/mL), and 22 months (SMA(22 M): 159.94 ± 21.71 pg/mL) (Figure 3A).

In parallel with the CSF results, when using the more precise SIMOA technology to measure total tau levels in plasma samples, the linear regression model adjusted for age and sex revealed significantly higher levels in the healthy control group (control: 7.51 ± 0.73 pg/mL) compared to plasma samples of non-treated adult SMA patients (SMA(Pre): 2.98 ± 0.37 pg/mL) [F(1,54) = 11.463, *p* = 0.001]. Additionally, the linear model results did not indicate age- [F(1,54) = 0.131, *p* = 0.719] or sex- [F(1,54) = 0.916, *p* = 0.343] dependent effects (Figure 3B).

Next, a one-way ANOVA of repeated measures did not reveal significant changes among plasma samples of adult SMA patients regardless of the treatment time point [F(2.7,61.43) = 1.056, *p* = 0.369]: non-treated adult SMA patients (SMA(Pre): (2.98 ± 0.37 pg/mL); and adult SMA patients treated at 2 months (SMA(2 M): 2.88 ± 0.31 pg/mL), 6 months (SMA(6 M): 2.87 ± 0.36 pg/mL), 10 months (SMA(10 M): 2.86 ± 0.37 pg/mL), and 22 months (SMA(22 M): 3.29 ± 0.48 pg/mL) (Figure 3B).

### 2.4. YKL-40 Levels in CSF and Plasma of Adult SMA Patients Treated with Nusinersen

The regression model, adjusted for age and sex, analyzing YKL-40 (also known as Chitinase 3-like 1) levels in CSF, revealed no significant differences between CSF samples of the healthy control group (control: 173.32 ± 11.39 ng/mL) and the non-treated adult SMA patient samples (SMA(Pre): 88.16 ± 6.50 ng/mL) related to the group effect [F(1,54) = 0.243, *p* = 0.624]. Furthermore, this analysis indicated a significant increase in YKL-40 CSF levels associated with age as a covariate [F(1,54) = 18.439, *p* = 0.000] but not with sex as a covariate [F(1,54) = 0.384, *p* = 0.538] (Figure 4A).

A subsequent one-way ANOVA of repeated measures did not reveal any significant changes in YKL-40 levels in CSF samples among adult SMA patients, regardless of the treatment and treatment time point. This includes non-treated adult SMA patients (SMA(Pre): 88.16 ± 6.50 ng/mL) as well as treated patients at different treatment time points: 2 months (SMA(2 M): 91.11 ± 2.64 ng/mL), 6 months (SMA(6 M): 93.89 ± 7.54 ng/mL), 10 months (SMA(10 M): 95.39 ± 5.81 ng/mL), and 22 months (SMA(22 M): 90.53 ± 5.00 ng/mL) (Figure 4A).

The YKL-40 results obtained in plasma samples showed a similar pattern to the observations in CSF. The regression model adjusted for age and sex, analyzing YKL-40 levels in plasma, did not reveal significant differences between the control group (control: 173.32 ± 11.39 ng/mL) and the non-treated adult SMA patient samples (SMA(Pre): 88.16 ± 6.50 ng/mL) due to the group effect [F(1,55) = 1.474, *p* = 0.230]. In this case, changes in plasma YKL-40 levels were not associated with age [F(1,55) = 0.896, *p* = 0.348] or sex as covariates [F(1,55) = 0.030, *p* = 0.864] (Figure 4B).

Next, a one-way ANOVA of repeated measures revealed a significant decrease in YKL-40 plasma levels in nusinersen-treated adult SMA patients at different treatment time points compared to non-treated adult SMA patients [F(2.93,69.59) = 3.212, *p* = 0.03]. Specifically, significant reductions were found in samples of adult SMA patients treated at 22 months (SMA(22 M): 30.01 ± 2.66 ng/mL, *p* = 0.01) when compared to non-treated patients (SMA(Pre): 37.44 ± 2.07 ng/mL, *p* = 0.01) and after 6 months of treatment (SMA(6 M): 38.12 ± 2.45 ng/mL, *p* = 0.002). However, no significant differences were observed among the other treatment time points, 2 months (SMA(2 M): 35.94 ± 2.64 ng/mL) and 10 months (SMA(10 M): 31.69 ± 3.48 ng/mL) (Figure 4B).

### 2.5. sAPPβ and Aβ40 Levels in CSF of Adult SMA Patients Treated with Nusinersen

Biomarkers derived from the APP processing pathway were analyzed in CSF as possible candidates in the context of adult SMA. The regression model, adjusted for age and sex, analyzing sAPPβ (soluble Amyloid Precursor Protein β) levels in CSF, did not reveal significant differences between the healthy control samples (control: 255.86 ± 12.63 pg/mL) and the non-treated adult SMA patient samples (SMA(Pre): 251.35 ± 15.42 pg/mL) due to the group effect [F(1,56) = 1.702, *p* = 0.198], not due to age [F(1,56) = 3.102, *p* = 0.084] or sex [F(1,56) = 0.063, *p* = 0.802] covariates.

A subsequent one-way ANOVA of repeated measures revealed significant and robust differences when comparing sAPPβ CSF levels of the non-treated adult SMA patients (SMA(Pre): 251.35 ± 15.42 pg/mL) with adult SMA patients treated at different time points [F(2.2,49.9) = 28.82, *p* = 0.000]: 2 months (SMA(2 M): 185.10 ± 13.32 pg/mL, *p* = 0.000), 6 months (SMA(6 M): 185.25 ± 15.84 pg/mL, *p* = 0.000), 10 months (SMA(10 M): 171.88 ± 15.67 pg/mL, *p* = 0.000), and 22 months (SMA(22 M): 169.08 ± 16.81 pg/mL, *p* = 0.000) (Figure 5A).

Similarly, the levels of Aβ40 and Aβ42 in CSF samples were determined using the ELISA test as products of the amyloidogenic pathway. The regression model adjusted for age and sex, analyzing Aβ40 levels in CSF, revealed a significant increase in the CSF baseline levels of adult SMA patients not treated with nusinersen (SMA(Pre): 2872.31 ± 285.77 pg/mL) compared to healthy control CSF levels (control: 2335.80 ± 260.72 pg/mL) due to the group effect [F(1,53) = 17.194, *p* = 0.000]. Furthermore, this analysis indicated that these significant changes in the CSF Aβ40 levels were also associated with age [F(1,53) = 17.365, *p* = 0.000] but not with sex as a covariate [F(1,53) = 0.683, *p* = 0.413] (Figure 5B).

Following this, the one-way ANOVA of repeated measures did not reveal significant differences when comparing the samples of non-treated adult SMA patients (SMA(Pre): 2872.31 ± 285.77 pg/mL) with those of adult SMA patients treated at different time points [F(2.686,60.43) = 2.113, *p* = 0.1143]: at 2 months (SMA(2 M): 2615.91 ± 249.31 pg/mL), 6 months (SMA(6 M): 2433.71 ± 243.85 pg/mL), 10 months (SMA(10 M): 2204.79 ± 243.18 pg/mL), and 22 months (SMA(22 M): 2389.21 ± 268.44 pg/mL). Aβ42 levels were not detected in most of the samples, only in a few cases, which did not allow for analysis.

### 2.6. Biomarkers’ Correlations: Biochemical and Clinical Correlations

Only 24 adult SMA patients were included in this subanalysis (see Section 4). Moreover, only sAPPβ levels were assessed, due to their significant variations after the treatment onset. To examine the association of sAPPβ with each clinical variable over time, a multivariate linear mixed regression was performed, accounting for the random effect of each patient. The analysis included as covariables the time point, the age at baseline, and the functional group (non-sitter, walker, sitter) to account for the variability in scale scores across different functional states. No significant association was observed between changes in sAPPβ levels and the functional group clinical scales scores of SMA patients. Results are summarized in Table 3.

## 3. Discussion

The objective of this study was to evaluate the potential of specific molecules as biomarkers of disease progression and of the response to nusinersen treatment in patients with SMA types 2 and 3. A total of thirty-one patients were recruited from the neuromuscular reference units of Bellvitge University Hospital in L’Hospitalet de Llobregat and La Fe University Hospital in Valencia. The sample size, although small, was considered suitable because the phenotype of SMA is heterogenous, ranging from severe to mild phenotypes, and the prevalence is low. The global prevalence is 1–2/100,000 inhabitants, and an incidence around 1 in 10,000 live births has been estimated, with SMA type I accounting for around 60% of all cases. The prevalence of both SMA type II and III together has been estimated to be around 1.5 per 100,000 [21].

Similarly to other studies [22,23], our data show that nusinersen treatment is associated with some improvements in adult SMA patients in clinical outcomes. In our study, “walkers” were those showing a greater (and statistically significant) response, followed by sitters (with a non-statistically significant improvement) and non-sitters (with no appreciable response to treatment). As previously suggested [11,23], most severely disease-affected patients are probably those with the most unfavorable risk–benefit ratio. These results are consistent with other previous studies that provide evidence of nusinersen efficacy in adult SMA, appearing to accumulate over time in individuals with walker and sitter forms of the disease. In contrast, in patients with extremely advanced disease, the effects on residual motor function are less clear [24]. These differences in improvement may reflect the relevance of physical activity and the presence of a larger pool of remaining motoneurons in response to disease treatment. Additionally, they may be related to the available scales, which may not be sensitive enough to detect changes in non-sitter patients [10]. Among the other potential reasons for the lack of improvement during the observational period of the study, it could also be that a longer observational period is necessary to observe changes, even if they are minor.

### 3.1. Neurofilament and Total Tau Protein Levels in Adult SMA Patients Treated with Nusinersen

Neurofilament proteins have been extensively studied as potential biomarkers in various conditions characterized by axonal injury and degeneration. Elevated levels of NF-H (neurofilament heavy chain) have been detected in blood and/or CSF in neurodegenerative disorders [25,26,27,28,29,30,31]. Similarly, increased levels of NF-L have been observed in Alzheimer’s disease (AD), amyotrophic lateral sclerosis (ALS), multiple sclerosis (MS), SMA type I, Charcot–Marie–Tooth disease, and adult-onset leukoencephalopathy with axonal spheroids and pigmented glia [25,26,27,28,29,30,31]. In addition, in the context of SMA patients, it has been demonstrated that nusinersen normalizes the levels of the axonal damage marker NF-L and correlates with motor improvement in children with SMA. More recently, elevated blood levels of pNF-H have been identified in SMA type I, and these are decreased following nusinersen therapy [32].

These data suggest that neurofilaments may serve as a valuable biomarker for monitoring treatment response in children with SMA [33]. However, in adult SMA patients, other researchers did not find significant changes in NF-L, pNF-H, and/or NF-H levels in CSF or plasma following nusinersen administration [34,35,36]. Only one study reported significant changes in pNF-H levels associated with nusinersen administration in adult SMA patients, where pNF-H decreased significantly in CSF over a 22-month follow-up period. However, these changes did not correlate with clinical outcome measures [34].

Similarly, our results indicate that adult SMA patients have lower baseline levels of NF-L and pNF-H compared to controls, but these differences are attributed to other variables and do not suggest consistent changes after nusinersen treatment in either CSF or plasma. These findings confirm that neurofilaments are not suitable for monitoring the disease progression or response to nusinersen treatment in adult patients with SMA types 2 and 3, only reflecting the already-demonstrated increase with age [33]. Regarding pNF-H, it is important to note that the applied methodology detected low amounts of pNF-H and showed a high degree of variability, indicating weak potential as a biomarker for this condition.

The microtubule-associated protein tau is expressed in neurons and plays a crucial role in axonal maintenance and transport. The rationale behind using tau as a biomarker for neurodegenerative diseases is that it is released by damaged cells. Consequently, elevated levels of tau in CSF serve as indicators of nerve cell damage in various neurodegenerative disorders [37,38]. In CSF, we found increased levels of tau in untreated adult SMA patients compared with healthy controls. These changes may indicate that neuronal death progresses in adult forms in a slow yet relentless manner, resulting in an increase in total tau levels within the pathological context. This increase is directly reflected in CSF but not in plasma, as there have been no reports of these changes in the latter. This lack of correlation between biofluids is not unique to our results; other studies have also observed similar patterns in pathologies characterized by increased neuronal death, suggesting that blood total tau primarily originates from peripheral, non-brain sources [39,40,41,42]. In plasma, differences in total tau levels were associated with age, as other researchers have reported previously [43].

In addition, our results indicated that there were no significant differences in total tau levels in CSF after the administration of several nusinersen doses. This lack of difference was observed in both CSF and plasma samples. Additionally, our findings are consistent with a previous study conducted on a small cohort of adult patients with SMA type 3 (n = 11), which found no changes in tau protein, S100B protein, and neuron-specific enolase in CSF after the loading phase of nusinersen [35]. Another study, involving nine adult patients with SMA type 2–3, analyzed CSF neurofilament light chain, total tau, phospho-tau, and serum creatinine levels. Neither changes in these biomarkers during the follow-up period were observed nor any correlation with motor scores at each time point were found [44]. Conversely, a study of SMA type 1 patients showed a reduction in both total CSF tau and NF-L with nusinersen administration [32]. Interestingly, another study in a heterogeneous population of SMA type 1–3 patients younger than 18 years old found a decrease in total tau (but not in NF-L) levels in CSF [45], suggesting that tau CSF levels might be more useful than NF-L to measure nusinersen response in pediatric forms and older children.

### 3.2. YKL-40 Protein Levels in CSF and Plasma in Adult SMA Patients Treated with Nusinersen

YKL-40 is a glycoprotein produced by inflammatory, cancer, and stem cells. Its physiological role is not completely understood, but YKL-40 levels are elevated in the brain and CSF in various neurological and neurodegenerative diseases characterized by increased inflammatory responses such as dementias and ALS [46,47]. Therefore, YKL-40 quantification may be useful in the evaluation of astrocytic responses in SMA types 2, 3, and 4 following nusinersen treatment.

Our results did not demonstrate consistent changes in YKL-40 levels with nusinersen administration in CSF, which differs from the findings of a single study that found a significant decrease at 22 months of treatment in patients showing improvement in the RULM score [34]. However, our findings only confirmed the increase in YKL-40 levels in CSF with aging [46]. In contrast, our plasma results indicate a significant reduction in YKL-40 plasma levels at the 22-month time point, consistent with observations in CSF from a previous study [34]. Our serum results must be treated with caution, as the changes have only been found at one time point, without correlating with CSF levels or showing a consistent downward trend. Additionally, it should be noted that the serum levels of YKL-40 can be influenced by systemic alterations that could interfere with its measurement, not directly reflecting what happens in the central nervous system [48,49].

### 3.3. sAPPβ Protein Levels Decrease after Treatment Onset but Do Not Associate with the Clinical Outcome

The proteolytic processing of amyloid precursor protein (APP) by α- or β-secretase results in two soluble metabolites, sAPPα and sAPPβ, respectively. Recent studies showed that plasma levels of sAPPβ are decreased in AD [50] and decreased in pathologies included along the ALS-FTLD spectrum [51,52,53]. Thus, sAPPβ levels may be a potential biomarker in other neurodegenerative diseases such as adult SMA. Our data demonstrated, for the first time, a significant decrease in sAPPβ levels during nusinersen treatment administration, suggesting it as a potential biomarker for treatment response and/or administration. This reduction might result in a decrease in neuroinflammation and also prevent cell death by promoting SMN recovery. It is important to note that sAPPβ stimulates microglia through its N-terminal domain upstream of residue 444, exhibiting neurotoxic effects [54]. Moreover, in the apoptosis of peripheral neurons induced by deprivation of growth factors, sAPPβ binds to the DR6 receptor triggering cell death [55]. However, this finding must be taken with caution since our multivariate model did not reveal a significant relationship between the clinical outcome and sAPPβ levels. Thus, further studies are necessary to better understand its role as a response biomarker.

In the context of adult SMA, there are only a few studies that focus on the APP processing pathway. However, none of these studies specifically examine sAPPβ; instead, they specifically examine Aβ40 and Aβ42 levels. One of these studies, conducted on a small cohort of adult SMA patients (n = 8) treated with nusinersen, determined that there was a significant increase in CSF levels of Aβ42 after one year of treatment [56]. In contrast, another study conducted on a cohort of SMA patients treated with nusinersen reported no significant changes in CSF levels of Aβ42 and Aβ40 during a follow-up period of less than 300 days [57]. In our case, Aβ42 levels were not detectable in most of the samples, and Aβ40 levels did not show significant changes during the treatment administration, although the latter were significantly increased in not-treated adult SMA patients compared to controls.

## 4. Materials and Methods

### 4.1. Study Design and SMA Patient Cohort

For this observational study, SMA patients from two reference centers in Spain were included (Hospital la Fe de Valencia, Hospital de Bellvitge). The inclusion criteria were as follows: (i) genetically confirmed SMA (either homozygous deletion or compound heterozygous mutation in SMN1); (ii) older than 18 years at the baseline visit; (iii) treated with nusinersen (at least five doses (6 months) at the time of the study closure, December 2022) following the protocol of the Health Department in Spain; and (iv) plasma and CSF samples available, at least, at baseline and at 6 months at the biobanks of Hospital de Bellvitge and Hospital la Fe de Valencia. Many of the patients included in this study were also included in a multicenter study that reported the clinical results of nusinersen treatment [11].

### 4.2. Cerebrospinal Fluid and Plasma Samples Collection and Nusinersen Administration

CSF (5 ± 0.5 mL) and plasma samples (4.0 ± 0.5 mL, using appropriate lithium heparin plasma separation tubes) were collected prospectively before each nusinersen dose as per the nusinersen label. Lumbar punctures were performed by experienced neurologists and neuroradiologists with local anesthesia. US/CT guidance was used, when needed, in patients with complex spines [58]. After obtaining the plasma and CSF samples, intrathecal nusinersen was administered following the label recommendations and schedule. CSF and plasma were centrifuged at 3000 rpm for 15 min at room temperature. Supernatant was collected and aliquoted in volumes of 250 μL and stored at −80 °C at the biobanks of Bellvitge and La Fe Hospitals until use. The extraction of biological samples was performed at the Functional Unit of Amyotrophic Lateral Sclerosis (UFELA) of the Neurology Service of the Bellvitge University Hospital. The CSF and plasma samples analyzed in this study were those obtained just before the first (time 0, SMA(Pre)), fourth (63 days, SMA(2 M)), fifth (6 months approx., SMA(6 M)), seventh (10 months approx., SMA(10 M)), and tenth (22 months approx., SMA(22 M)) nusinersen dose. Treatment was interrupted in those patients suffering from side effects deemed as intolerable by the patients and/or neurologists or in those patients with complex spines in whom lumbar access was no longer available. As controls, we used samples obtained from 30 patients without neurological conditions in the Traumatology Unit of the Bellvitge Hospital. All samples were analyzed after a single freeze/thaw cycle.

### 4.3. Clinical Data

Detailed clinical data were collected in all patients. Retrospective data included the age of onset, age at diagnosis, type of SMA, clinical symptoms, and genetic characteristics (including SMN2 copy number). Prospective data included the Hammersmith functional motor scale expanded (HFMSE) with video-recording, the revised upper limb module (RULM) with video-recording, the ALSFSR-R score, and the forced vital capacity (FVC). Clinical data were collected right before the first nusinersen dose and then every 12 months, approximately.

### 4.4. Biomarker Analysis in CSF and Plasma Samples Using ELISA Kits

CSFs from adult SMA patients were analyzed, in parallel with samples from healthy non-age-matched control patients, for neurodegeneration biomarkers using the appropriate ELISA kit test. Total tau (t-Tau) protein was quantified using the INNOTEST hTAU AG ELISA kit from Fujirebio (Cat nº 81572, Fujirebio, Les Ulis, France); pNF-H (phosphorylated neurofilament heavy) levels were quantified using the phosphorylated Neurofilament heavy (pNF-H) Sandwich ELISA kit from MerckMillipore (Cat nº NS170, MerckMillipore, Burlington, MA, USA); NFL (neurofilament light) levels were detected using the NF-Light ELISA Assay kit from UmanDiagnostics (Cat nº 10-7001, UmanDiagnostics, Umea, Sweden); sAPPβ (soluble Amyloid Precursor Protein Beta) was determined using the sAPP-β wild type high sensitive ELISA kit from Creative Diagnostic (Cat nº DEIA6295, Creative Diagnostic, Upton, NY, USA), and YKL-40 was analyzed using MicroVue YKL-40 EIA from Quidel (Cat nº 8020, Quidel, San Diego, CA, USA). YKL-40 levels in plasma samples were also evaluated using the same kit MicroVue YKL-40 EIA from Quidel (Cat nº 8020, San Diego, Quidel, CA, USA).

### 4.5. Biomarker Analysis in Plasma Samples Using SIMOA Technology

Plasma from adult SMA patients were analyzed, in parallel with samples from healthy non-age-matched control patients, for neurodegeneration biomarkers using SIMOA technology to detect plasma levels of total tau and NF-L. All plasma samples were processed at the end of the project, including samples from the first to second year dose treatment (five time dose samples included) due to methodology procedures.

### 4.6. Statistical Analysis

The normality of distribution was analyzed with the Kolmogorov–Smirnov test. The evolution of clinical outcome measures at the different time points has been described in tables, and the means have been represented graphically. The Wilcoxon matched-pairs signed-rank test was used to compare baseline clinical score values with intermediate and final scores. Given that biomarkers did not follow a normal distribution and there were significant age differences between controls and SMA patients, a linear regression approach was applied to determine the differences in biomarker levels between pretreatment patients and controls, taking age and sex into account as covariables [59]. The data were presented in a box plot, with the first quartile to the third quartile represented by the box and a horizontal line indicating the median. Significance levels between compared groups and studied covariables were represented and set as follows: for the group effect, $ *p* < 0.05, $$ *p* < 0.01, and $$$ *p* < 0.001; for the age effect, # *p* < 0.05, ## *p* < 0.01, and ### *p* < 0.001; and for the sex effect, & *p* < 0.05, && *p* < 0.01, and &&& *p* < 0.001.

Next, to investigate the modulation of biomarkers during treatment administration, the statistical analysis of the biomarker level data between groups was conducted using a one-way repeated measures ANOVA (mixed model) followed by a Tukey post-test. This analysis was performed using the SPSS software (IBM Corp. Released 2013. IBM SPSS Statistics for Windows, Version 21.0. Armonk, NY: IBM Corp.). Outliers were detected using the GraphPad software QuickCalcs (*p* < 0.05). Graphic design was performed with GraphPad Prism version 9.05 (La Jolla, CA, USA). The data were presented in a box plot, with the first quartile to the third quartile represented by the box and a horizontal line indicating the median. Significance levels between compared groups during treatment administration were set at * *p* < 0.05, ** *p* < 0.01, and *** *p* < 0.001.

The association over time of the different biomarkers with each clinical variable was assessed with a multivariate linear mixed regression, taking into account the random effect of the patient. Adjusting variables were the time point, the age at baseline, and the interaction with the functional state (non-sitter, walker, sitter). The conditions of application of the models have been validated, and confidence intervals at 95% of the estimators have been calculated whenever possible. All analyses were conducted with the statistic program R version 4.1.0 (18 May 2021) for Windows.

## 5. Conclusions

SMA patients demonstrated limited improvements in clinical outcomes following nusinersen treatment, with statistically significant effects observed primarily in ‘walker’ patients but not in those severely affected (‘sitter’ and ‘non-sitter’). The slower disease progression and mild response to treatment in adult SMA types 2 and 3 did not result in detectable changes in common markers of axonal degradation, inflammation, or neurodegeneration, as seen in pediatric forms where larger pools of damaged neurons are present. Specifically, the neurodegeneration biomarkers investigated in this study (NF-L, pNF-H, total tau, YKL-40, Aβ40, and Aβ42) did not consistently show significant changes with nusinersen treatment. This lack of significant changes aligns with previous findings from similar studies conducted in the context of adult SMA. However, a reduction in sAPPβ levels was noted for the first time following nusinersen treatment. However, despite this reduction, these levels did not correlate with clinical outcomes and remained stable during treatment administration.

All these findings suggest a potential link between changes in sAPPβ levels and treatment administration, warranting further investigation to better understand this phenomenon. Additionally, our data underscore the need for discovery studies of novel molecules tailored to the specific pathological context of adult SMA and available treatments, as canonical neurodegenerative biomarkers are inadequate for this purpose. This approach will allow for a much more precise prediction and study of treatment response at a molecular level, moving away from conventional methods.

## Figures and Tables

**Figure 1 ijms-25-03810-f001:**
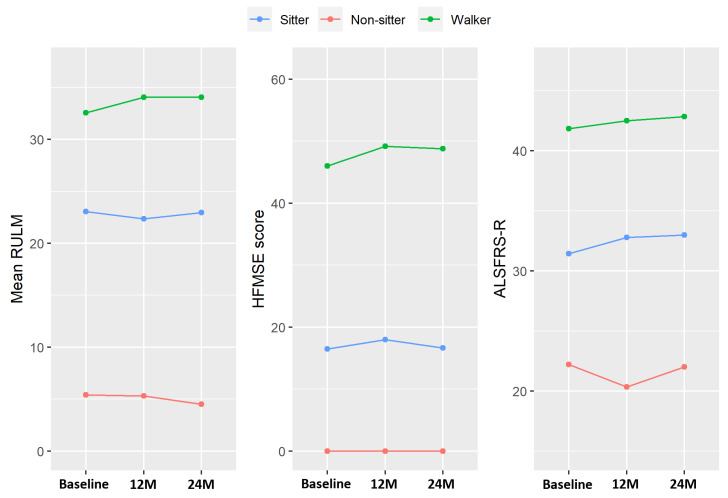
A graphic representation of the clinical evolution data for SMA patients undergoing nusinersen treatment, stratified by non-sitters, sitters, and walkers. The data are presented for the RULM, HFMSE, and ALS-FRS-R scales at baseline, 12 months, and 24 months. The graph illustrates the changes in scores over time for each group, providing a visual representation of the treatment’s impact on the different functional measures.

**Figure 2 ijms-25-03810-f002:**
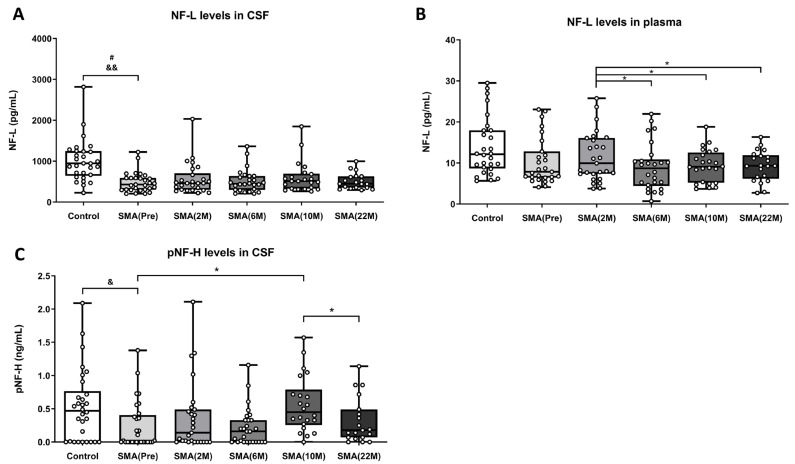
(**A**) Levels of NF-L (pg/mL) measured in the CSF of control subjects (n = 29), nusinersen non-treated adult SMA patients (Pre) (n = 29), and nusinersen-treated adult SMA patients at 2-month (2 M) (n = 26), 6-month (6 M) (n = 26), 10-month (10 M) (n = 22), and 22-month (22 M) (n = 19) time points. (**B**) Levels of NF-L (pg/mL) measured in the plasma of control subjects (n = 29), nusinersen non-treated adult SMA patients (Pre) (n = 27), and nusinersen-treated adult SMA patients at 2-month (2 M) (n = 27), 6-month (6 M) (n = 26), 10-month (10 M) (n = 24), and 22-month (22 M) (n = 21) time points. (**C**) Levels of pNF-H (ng/mL) measured in the CSF of control subjects (n = 29), nusinersen non-treated adult SMA patients (Pre) (n = 29), and nusinersen-treated adult SMA patients at 2-month (2 M) (n = 26), 6-month (6 M) (n = 28), 10-month (10 M) (n = 22), and 22-month (22 M) (n = 19) time points. The data are presented in a box plot, with the first quartile to the third quartile represented by the box and a horizontal line indicating the median. The whiskers extend from each quartile to the minimum or maximum values. Significance levels between the control and non-treated adult SMA patient groups, using the linear regression model while considering covariables, are represented and set as follows: for the age effect, # *p* < 0.05, and for the sex effect, & *p* < 0.05 and && *p* < 0.01. Statistical significance levels in paired group comparisons during treatment are indicated as * *p* < 0.05. Abbreviations: M: month.

**Figure 3 ijms-25-03810-f003:**
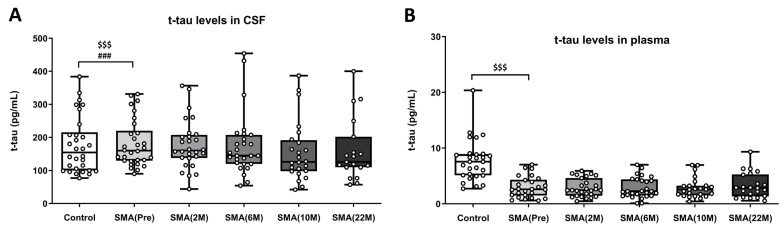
(**A**) Levels of total tau (pg/mL) measured in the CSF of control subjects (n = 30), nusinersen non-treated adult SMA patients (Pre) (n = 30), and nusinersen-treated adult SMA patients at 2-month (2 M) (n = 28), 6-month (6 M) (n = 27), 10-month (10 M) (n = 24), and 22-month (22 M) (n = 19) time points. (**B**) Levels of total tau (pg/mL) measured in the plasma of control subjects (n = 28), nusinersen non-treated adult SMA patients (Pre) (n = 26), and nusinersen-treated adult SMA patients at 2-month (2 M) (n = 27), 6-month (6 M) (n = 25), 10-month (10 M) (n = 24), and 22-month (22 M) (n = 22) time points. The data are presented in a box plot, with the first quartile to the third quartile represented by the box and a horizontal line indicating the median. The whiskers extend from each quartile to the minimum or maximum values. Significance levels between the control and non-treated adult SMA patient groups, using the linear regression model while considering covariables, are represented and set as follows: for the group effect, $$$ *p* < 0.001, and for the age effect, ### *p* < 0.001. Statistical significance was not found in paired group comparisons during treatment. Abbreviations: M: month.

**Figure 4 ijms-25-03810-f004:**
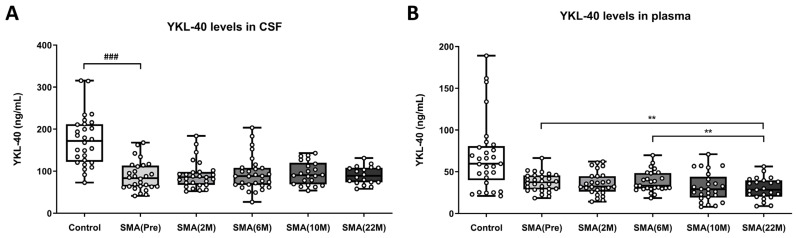
(**A**) Levels of YKL-40 (ng/mL) measured in the CSF of control subjects (n = 28), nusinersen non-treated adult SMA patients (Pre) (n = 30), and nusinersen-treated adult SMA patients at 2-month (2 M) (n = 28), 6-month (6 M) (n = 27), 10-month (10 M) (n = 24), and 22-month (22 M) (n = 19) time points. (**B**) Levels of YKL-40 (ng/mL) measured in the plasma of control subjects (n = 29), nusinersen non-treated adult SMA patients (Pre) (n = 28), and nusinersen-treated adult SMA patients at 2-month (2 M) (n = 27), 6-month (6 M) (n = 29), 10-month (10 M) (n = 25), and 22-month (22 M) (n = 23) time points. The data are presented in a box plot, with the first quartile to the third quartile represented by the box and a horizontal line indicating the median. The whiskers extend from each quartile to the minimum or maximum values. Significance levels between the control and non-treated adult SMA patient groups, using the linear regression model while considering covariables, are represented and set as follows: for the age effect, ### *p* < 0.001. Statistical significance levels in paired group comparisons during treatment are indicated as ** *p* < 0.01. Abbreviations: M: month.

**Figure 5 ijms-25-03810-f005:**
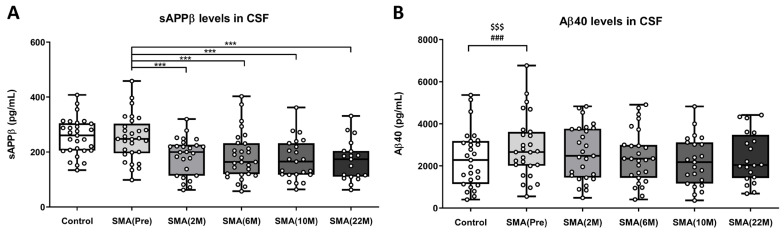
(**A**) Levels of sAPPβ (pg/mL) measured in the CSF of control subjects (n = 29), nusinersen non-treated adult SMA patients (Pre) (n = 29), and nusinersen-treated adult SMA patients at 2-month (2 M) (n = 26), 6-month (6 M) (n = 28), 10-month (10 M) (n = 23), and 22-month (22 M) (n = 19) time points. (**B**) Levels of Aβ40 (pg/mL) measured in the CSF of control subjects (n = 27), nusinersen non-treated adult SMA patients (Pre) (n = 27), and nusinersen-treated adult SMA patients at 2-month (2 M) (n = 27), 6-month (6 M) (n = 27), 10-month (10 M) (n = 23), and 22-month (22 M) (n = 20) time points. The data are presented in a box plot, with the first quartile to the third quartile represented by the box and a horizontal line indicating the median. The whiskers extend from each quartile to the minimum or maximum values. Significance levels between the control and non-treated adult SMA patient groups, using the linear regression model while considering covariables, are represented and set as follows: for the group effect, $$$ *p* < 0.001, and for the age effect, ### *p* < 0.001. Statistical significance levels in paired group comparisons during treatment are indicated as *** *p* < 0.001. Abbreviations: M: month.

**Table 1 ijms-25-03810-t001:** Baseline characteristics of SMA patients included in biochemical and clinical analysis.

	Patients Included in the Baseline Biomarker Analysis (n = 31)	Patients Included in the Longitudinal Analysis (n = 24)
Age, Mean (SD)	37.3 (2.20)	35.13 (10.37)
Missing data	1	1
Sex, n (%)		
Man	19 (61.29%)	17 (70.83%)
Woman	12 (38.71%)	7 (29.17%)
SMA type, n (%)		
II	7 (22.58%)	5 (20.83%)
III	24 (77.42%)	19 (79.17%)
SMA subtype, n (%)		
2a	5 (16.14%)	4 (16.67%)
2b	1 (3.23%)	-
3a	9 (29.00%)	9 (37.50%)
3b	16 (51.63%)	11 (45.83%)
Functional state (walker, sitter, non-sitter), n (%)		
Non-sitter	8 (25.80%)	5 (20.83%)
Sitter	12 (38.70%)	11 (45.83%)
Walker	11 (35.50%)	8 (33.33%)

**Table 2 ijms-25-03810-t002:** A description of the clinic scales in function of the visit.

	Baseline, N = 24	12 Months, N = 24	24 Months, N = 24
Mean RULM			
Mean (SD)	22.54 (11.86)	22.72 (12.23)	23.27 (12.35)
Median (IQR)	26.25 (14.00, 32.25)	25.00 (14.00, 34.75)	23.50 (15.50, 36.50)
Missing values	0	1	0
HFMSE score			
Mean (SD)	22.88 (20.32)	24.10 (21.36)	25.61 (21.82)
Median (IQR)	18.50 (5.00, 35.25)	18.00 (2.00, 39.00)	23.00 (5.50, 45.50)
Missing values	0	3	1
ALSFRS-R			
Mean (SD)	32.25 (8.44)	34.80 (8.74)	35.76 (8.42)
Median (IQR)	32.00 (25.75, 40.25)	36.50 (29.25, 41.50)	40.00 (30.00, 42.00)
Missing values	4	4	7

**Table 3 ijms-25-03810-t003:** A linear mixed regression of the effect of sAPPβ levels in CSF with RULM, HFMSE, and ALS-FSR-R scores.

RULM	sAPPβ (pg/mL)—CSF		
Predictor	Estimates	CI	*p*-value
Scale (biomarker)—non-sitter	1.18	−3.00–5.36	0.572
Scale (biomarker)—sitter	1.10	−0.58–2.79	0.195
Scale (biomarker)—walker	−0.24	−2.28–1.79	0.810
HFMSE	sAPPβ (pg/mL)—CSF		
Predictor	Estimates	CI	*p*-value
Scale (biomarker)—non-sitter	0.24	−6.92–7.39	0.947
Scale (biomarker)—sitter	0.87	−2.10–3.83	0.559
Scale (biomarker)—walker	1.38	−1.99–4.75	0.414
ALS-FRS-R	sAPPβ (pg/mL)—CSF		
Predictor	Estimates	CI	*p*-value
Scale (biomarker)—non-sitter	2.64	−0.90–6.17	0.139
Scale (biomarker)—sitter	−0.37	−1.65–0.91	0.559
Scale (biomarker)—walker	−0.23	−1.74–1.28	0.760

## Data Availability

Data sharing is not applicable to this article as no datasets were generated or analyzed during this study.

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
