# Peer review of "Neurodegeneration Biomarkers in Adult Spinal Muscular Atrophy (SMA) Patients Treated with Nusinersen"

_ijms, 2024, doi:10.3390/ijms25073810_

Round 1
Reviewer 1 Report
Comments and Suggestions for Authors
The topic of the work is very interesting. In fact, this is a line of research that many international groups are carrying out both on neurofilaments and on other biomarkers.
The strengths that I see in the work are mainly two:
1) having collected and analyzed the biological fluids of both healthy patients and affected patients. So having the control group matched by age and sex.
2) having analyzed the values of the biomarkers at time zero corresponding to the period before the administration of drugs.
The main limitation of the research is represented by the heterogeneity of the sample and the minimum sample size.
The objective of these works is to enrich the international community's knowledge of these biomarkers.
The ultimate goal of knowing these biomarkers will be to use them to understand the clinical progress of patients and their different response to different therapeutic options.

Author Response
No modifications were made to our study based on the revision by Reviewer 1, as they had already approved the original version of the draft without any inquiries.
Reviewer 2 Report
Comments and Suggestions for Authors
I have reviewed the manuscript by Andrés-Benito et al., titled "Neurodegeneration Biomarkers in Adult Spinal Muscular Atrophy (SMA) Patients Treated with Nusinersen" submitted to the journal IJMS. Overall, the content and presentation are average, but some areas require revisions, without which it is not publishable:
The introduction lacks clarity regarding why the authors chose nusinersen over other molecules. It would be beneficial to provide a brief rationale or background for selecting nusinersen as the focus of the study in the introduction itself (maybe in the last paragraph).
The language throughout could be improved for clarity and coherence. There are instances of awkward phrasings and unclear sentences. A thorough proofreading and editing process is necessary to enhance readability.
Consider adding a concept diagram to visually summarize the key findings and take-home message from the study. This can help readers better understand the main results and implications of the research at a glance.
Expand the discussion of findings to provide more insight into the implications of the results. Discuss the significance of the observed changes in biomarkers and their correlation with clinical outcomes. Consider comparing the findings with previous studies in the field.
With these minor but essential revisions, the manuscript will be suitable for publication in the journal IJMS.
Comments on the Quality of English LanguageOK
Author Response
I have reviewed the manuscript by Andrés-Benito et al., titled "Neurodegeneration Biomarkers in Adult Spinal Muscular Atrophy (SMA) Patients Treated with Nusinersen" submitted to the journal IJMS. Overall, the content and presentation are average, but some areas require revisions, without which it is not publishable:
Q: The introduction lacks clarity regarding why the authors chose nusinersen over other molecules. It would be beneficial to provide a brief rationale or background for selecting nusinersen as the focus of the study in the introduction itself (maybe in the last paragraph).
R: “The present study only includes nusinersen as an adult SMA treatment because at the time of its design and approval (2020), it was the only drug approved at the state level in Spain by the “Spanish Agency of Medicines and Medical Devices” for the treatment of adult SMA.” It has been added in the introduction’s last paragraph and it has been indicated in green.
Q: The language throughout could be improved for clarity and coherence. There are instances of awkward phrasings and unclear sentences. A thorough proofreading and editing process is necessary to enhance readability.
R: The whole draft has been reviewed by an English editor; mistakes have been corrected. Language precision and refinements have been performed in the text to improve readability.
Q: Consider adding a concept diagram to visually summarize the key findings and take-home message from the study. This can help readers better understand the main results and implications of the research at a glance.
R: In response to another reviewer's request for a conclusion section, we have included this section instead of the diagram. This conclusion section also summarizes the key findings and the take-home message from the study.
Q: Expand the discussion of findings to provide more insight into the implications of the results. Discuss the significance of the observed changes in biomarkers and their correlation with clinical outcomes. Consider comparing the findings with previous studies in the field.
R: In the first part of the discussion, we included additional clinical studies. However, in the biomarkers section of the discussion, all similar works carried out in the field, including those performed in pediatric forms, were included, discussed, and compared, despite the fact that they were conducted on cohorts of patients smaller than ours. Since it is a rare disease, and especially its adult form, the number of studies available is limited to those presented here, making it difficult to expand the discussion due to this lack of studies. Find here the main existing studies, all of them discussed in the discussion section:
- Vázquez-Costa JF, Povedano M, Nascimiento-Osorio AE, Moreno Escribano A, Kapetanovic Garcia S, Dominguez R, et al. Nusinersen in adult patients with 5q spinal muscular atrophy: A multicenter observational cohorts’ study. Eur J Neurol. 2022 Nov;29(11):3337–46.
- Meyer T, Maier A, Uzelac Z, Hagenacker T, Günther R, Schreiber-Katz O, et al. Treatment expectations and perception of therapy in adult patients with spinal muscular atrophy receiving nusinersen. Eur J Neurol. 2021 Aug;28(8):2582–95.
- Hagenacker T, Wurster CD, Günther R, Schreiber-Katz O, Osmanovic A, Petri S, et al. Nusinersen in adults with 5q spinal muscular atrophy: a non-interventional, multicentre, observational cohort study. Lancet Neurol. 2020 Apr;19(4):317–25.
- Darras BT, Crawford TO, Finkel RS, Mercuri E, De Vivo DC, Oskoui M, et al. Neurofilament as a potential biomarker for spinal muscular atrophy. Ann Clin Transl Neurol. 2019 May;6(5):932–44.
- Olsson B, Alberg L, Cullen NC, Michael E, Wahlgren L, Kroksmark AK, et al. NFL is a marker of treatment response in children with SMA treated with nusinersen. J Neurol. 2019 Sep;266(9):2129–36.
- De Wel B, De Schaepdryver M, Poesen K, Claeys KG. Biochemical and clinical biomarkers in adult SMA 3–4 patients treated with nusinersen for 22 months. Ann Clin Transl Neurol. 2022 Aug;9(8):1241–51.
- Totzeck A, Stolte B, Kizina K, Bolz S, Schlag M, Thimm A, et al. Neurofilament heavy chain and tau protein are not elevated in cerebrospinal fluid of adult patients with spinal muscular atrophy during loading with nusinersen. Int J Mol Sci. 2019 Nov;20(21).
- Rich KA, Fox A, Yalvac M, Heintzman S, Tellez M, Bartlett A, et al. Neurofilament Levels in CSF and Serum in an Adult SMA Cohort Treated with Nusinersen. J Neuromuscul Dis. 2022;9(1):111–9.
- Milella G, Introna A, D’Errico E, Fraddosio A, Scaglione G, Morea A, et al. Cerebrospinal Fluid and Clinical Profiles in Adult Type 2–3 Spinal Muscular Atrophy Patients Treated with Nusinersen: An 18-Month Single-Centre Experience. Clin Drug Investig. 2021 Sep;41(9):775–84.
- Šimić G, Vukić V, Babić M, Banović M, Berečić I, Španić E, et al. Total tau in cerebrospinal fluid detects treatment responders among spinal muscular atrophy types 1–3 patients treated with nusinersen. CNS Neurosci Ther. 2024 Mar;30(3):e14051.
- Introna A, Milella G, D’Errico E, Fraddosio A, Scaglione G, Ucci M, et al. Is cerebrospinal fluid amyloid‐β42 a promising biomarker of response to nusinersen in adult spinal muscular atrophy patients? Muscle Nerve. 2021 Jun;63(6):905–9.
- Walter MC, Wenninger S, Thiele S, Stauber J, Hiebeler M, Greckl E, et al. Safety and Treatment Effects of Nusinersen in Longstanding Adult 5q-SMA Type 3 - A Prospective Observational Study. J Neuromuscul Dis. 2019;6(4):453–65.
Reviewer 3 Report
Comments and Suggestions for Authors
Spinal muscular atrophy is a serious rare disease in which, due to a genetic defect, neurons in the spinal cord responsible for muscle contraction and relaxation gradually die. The lack of nerve impulses means that the skeletal muscles are not stimulated, weaken and gradually atrophy, which may lead to partial or complete paralysis.
Until recently, SMA was an incurable disease leading to respiratory failure and severe motor disability. In recent years, there has been a breakthrough in the treatment of SMA. The condition for the effectiveness of SMA treatment is early initiation of therapy, before irreversible loss of motor neurons in the spinal cord. Three causal drugs have been registered, increasing the amount of the survival of motor neuron protein, the deficiency of which causes the disease, one of them is nusinersen. All the above-mentioned drugs are highly effective, but depend on the stage of the disease at the time of initiation of therapy. The best results are achieved by starting treatment at a pre-symptomatic stage. Therefore, testing for this disease is extremely important.
Explanations of abbreviations should be provided the first time they are used. The incorrect entry has the abbreviation CSF, the translation of which is only found in the methodology on line 432, and is used many times before. On horseback, a list of cows should be created.
Figure 2 I don't understand what the difference is in points A and C because both here and there the level of NF-L in the CSF is analyzed at the same time points.
What is control if SMA(pre) is patients before treatment?
Figures: What value does statistical significance refer to? Always for control or SMA(pre) or other samples? If it is always for inspection, the entry on the graphics does not indicate this and is misleading. And if not for control, on what basis do you choose the reference point?
The presented results do not show a spectacular impact of long-term therapy. Is it possible to improve its effectiveness? e.g. by using higher concentrations?
The publication lacks conclusions that would summarize the results.
The bibliography does not meet the editorial requirements of the Journal.
The article cannot be published without answers to questions and corrections.
Author Response
Spinal muscular atrophy is a serious rare disease in which, due to a genetic defect, neurons in the spinal cord responsible for muscle contraction and relaxation gradually die. The lack of nerve impulses means that the skeletal muscles are not stimulated, weaken and gradually atrophy, which may lead to partial or complete paralysis.
Until recently, SMA was an incurable disease leading to respiratory failure and severe motor disability. In recent years, there has been a breakthrough in the treatment of SMA. The condition for the effectiveness of SMA treatment is early initiation of therapy, before irreversible loss of motor neurons in the spinal cord. Three causal drugs have been registered, increasing the amount of the survival of motor neuron protein, the deficiency of which causes the disease, one of them is nusinersen. All the above-mentioned drugs are highly effective, but depend on the stage of the disease at the time of initiation of therapy. The best results are achieved by starting treatment at a pre-symptomatic stage. Therefore, testing for this disease is extremely important.
Q: Explanations of abbreviations should be provided the first time they are used. The incorrect entry has the abbreviation CSF, the translation of which is only found in the methodology on line 432, and is used many times before. On horseback, a list of cows should be created.
R: It has been adapted and revised based on APA abbreviation recommendations in the main text. As to the last part of the question, none of us can make sense of the last sentence in this paragraph, "On horseback, a list of cows should be created." We assume that it must refer to something related to the abbreviations, but all of them have been revised.
Q: Figure 2 I don't understand what the difference is in points A and C because both here and there the level of NF-L in the CSF is analyzed at the same time points.
R: The difference remains in that A) Figure refers to NF-L levels in CSF and C) refers to pNF-H. These are two different molecules usually used to assess neurodegenerative conditions. This was indicated in the first version of the draft.
Q: What is control if SMA(pre) is patients before treatment?
R: As it was explained in the first draft, the current line nº454 (aprox.), “As controls, we used samples obtained from 30 patients without neurological conditions in the Traumatology Unit of the Bellvitge Hospital.” This group act as a disease control group, and the group SMA(pre) serves as control group for treatment administration (baseline).
Q: Figures: What value does statistical significance refer to? Always for control or SMA(pre) or other samples? If it is always for inspection, the entry on the graphics does not indicate this and is misleading. And if not for control, on what basis do you choose the reference point?
R: In the initial version of the draft, we already indicated significant differences between groups on each figure using a bar. Simultaneously, the main text (results section) specifies the comparisons to which these differences belong.
Q: The presented results do not show a spectacular impact of long-term therapy. Is it possible to improve its effectiveness? e.g. by using higher concentrations?
R: At the time of this study (2020-2022), nusinersen was administered according to the protocol used in this project, at the recommended concentration provided by the manufacturer, and with the specified time-dose administration. As of now, this nusinersen treatment protocol remains the only available option until further studies with higher doses are accepted. Currently, there is an ongoing study investigating higher doses of nusinersen in pediatric forms of the disease, aiming to enhance treatment response. However, it's important to note that these higher doses are not yet approved, are still under investigation and are not addressed for adult forms [Finkel RS, Day JW, Pascual Pascual SI, Ryan MM, Mercuri E, De Vivo DC, Montes J, Gurgel-Giannetti J, Monine M, Gambino G, Makepeace C, Foster R, Berger Z; DEVOTE Study Group. DEVOTE Study Exploring Higher Dose of Nusinersen in Spinal Muscular Atrophy: Study Design and Part A Results. J Neuromuscul Dis. 2023;10(5):813-823. doi: 10.3233/JND-221667].
Q: The publication lacks conclusions that would summarize the results.
R: Conclusions section has been included and indicated in green.
Q: The bibliography does not meet the editorial requirements of the Journal.
R: We have removed the DOI number from the references that already include it and adjusted the book references to comply with the editorial instructions.
Round 2
Reviewer 3 Report
Comments and Suggestions for Authors
Sorry for the incomprehensible part, there was a mistake. I wanted to create a list of abbreviations at the end of the publication. But the improvements you made are ok and I have no further comments.
I understand the comment about statistical significance, but I believe that the statistics should be presented in the figure in such a way that it does not raise any doubts, without the need to read the text. The text should only be an analysis and interpretation of the results presented in the figure.
If you are actually comparing different groups, the figure is ok. I asked to make sure.
In the case of the bibliography, it should be further improved:
- Journal titles in italics
- the year should be bold
- DOI is required and its removal was unnecessary
- the month of publication should also be removed
Epample:
Chen, Y.; Gao, Y.; Li, Y.; Yin, J. Anti-Biofilm Activity of Assamsaponin A, Theasaponin E1, and Theasaponin E2 against Candida albicans. Int. J. Mol. Sci. 2024, 25, 3599. https://doi.org/10.3390/ijms25073599
Thank you for following the rest of my opinions and I believe that after correcting the bibliography, the article is suitable for publication.
Author Response
Dear Reviewer,
Based on your previous comments, we have readjusted the figures to present the statistical results in a clearer manner, without the need to refer to the text. Additionally, we have added legends for the statistics. Changes are indicated in green. We believe that this adjustment enhances the readability of the results.
We have also modified the references to adhere to the journal's style. We apologize for any inconvenience caused by this oversight.
Thank you for your valuable feedback.